# Infrared Imaging Analysis of Green Composite Materials during Inline Quasi-Static Flexural Test: Monitoring by Passive and Active Approaches

**DOI:** 10.3390/ma16083081

**Published:** 2023-04-13

**Authors:** Massimo Rippa, Vito Pagliarulo, Francesco Napolitano, Teodoro Valente, Pietro Russo

**Affiliations:** 1Institute of Applied Sciences and Intelligent Systems “E. Caianiello”, National Research Council, 80078 Pozzuoli, NA, Italy; 2Institute for Polymers, Composites and Biomaterials, National Research Council, 80078 Pozzuoli, NA, Italy; 3Department of Chemical Engineering Materials Environment and UdR INSTM, Sapienza-Università di Roma, 00185 Rome, Italy

**Keywords:** infrared imaging, composite materials, thermography, inline test, jute, basalt

## Abstract

Composite materials have been used for many years in a wide variety of sectors starting from aerospace and nautical up to more commonly used uses such as bicycles, glasses, and so on. The characteristics that have made these materials popular are mainly their low weight, resistance to fatigue, and corrosion. In contrast to the advantages, however, it should be noted that the manufacturing processes of composite materials are not eco-friendly, and their disposal is rather difficult. For these reasons, in recent decades, the use of natural fibers has gained increasing attention, allowing the development of new materials sharing the same advantages with conventional composite systems while respecting the environment. In this work, the behavior of totally eco-friendly composite materials during flexural tests has been studied through infrared (IR) analysis. IR imaging is a well-known non-contact technique and represents a reliable means of providing low-cost in situ analysis. According to this method, the surface of the sample under investigation is monitored, under natural conditions or after heating, by recording thermal images with an appropriate IR camera. Here, the results achieved for jute- and basalt-based eco-friendly composites through the use of both passive and active IR imaging approaches are reported and discussed, showing the possibilities of use also in an industrial environment.

## 1. Introduction

In recent years, the growing sensitivity of public opinion towards a more sustainable world has prompted research, both industrial and academic, to study the potential offered by natural fiber-based composite materials, effectively creating the first practical applications of these materials, especially in the naval and automotive fields. The characteristics of natural fiber composites are very close to those of traditional composites (i.e., carbon fiber and glass fiber), being lightweight, high strength, high modulus, fatigue resistant, and corrosion resistant. In addition, their use makes it possible to reduce the pollution associated with the production and disposal of the traditional ones, becoming their perfect substitutes [1,2,3,4,5,6]. However, the most ambitious goal is to have a completely eco-friendly composite material, i.e., fibers and matrices, which can completely replace polluted materials. This is not a simple task because natural fibers are intrinsically hydrophilic and typically incompatible with most host polymeric matrices that have a hydrophobic nature. These factors lead to a poor interfacial adhesion, which, in turn, determines the limited mechanical performance of the obtained materials, with the triggering, during their useful life, of disruptive phenomena such as delamination and debonding, even under low external loads. This greatly limits the choice of matrix. The development of these materials passes through the execution of tests and analyzes useful for the characterization of their properties and their performances. In this context, the use of an eco-sustainable matrix used together with natural fibers would make it possible to create a completely eco-friendly composite material. It is, therefore, important to have the tools capable of characterizing these new materials in a clear and rapid way and to establish their performance. The development of new investigation techniques that can be implemented in situ and managed remotely can help to better understand the mechanical performance and the limits of the new materials designed by suggesting possible improvements.

Looking at the wide world of diagnostic tools, imaging techniques have become widely applied for materials characterizations providing structural information of the samples useful to take decisions in their development. Infrared (IR) imaging is a well-known non-contact, non-invasive technique. Compared to other non-destructive methods, it can allow a low-cost in situ investigation, providing information on the state of health of the investigated sample in a short time and allows inspection even when only one side of it is accessible [7,8,9,10,11]. According to this method, the surface of the sample under investigation is monitored, under natural conditions or after heating using an external source, by recording thermal images with an appropriate infrared camera. It represents an efficient approach of analysis applied in many fields, among which are: the aerospace, engineering, cultural heritage, agriculture, and new materials investigations [12,13,14,15,16,17,18,19]. In these last cases, IR imaging is effective in detecting the presence of a wide variety of both surface and sub-surface defects or damages such as inclusions, voids, cracks, detachments, delaminations, and any type of structural inhomogeneities that determines a change in the thermo-physical properties of the sample under investigation [20,21,22,23,24,25,26]. Here, we report on infrared imaging analysis performed during inline quasi-static flexural test of composite laminates based on a commercial eco-friendly blend matrix reinforced with two of the most used natural fibers, jute and basalt [27,28,29]. Passive and active approaches were used to compare the thermal response of the materials and to detect the damages induced during the test. In the case of active analysis, performed by a pulsed thermal stimulation, we calculate and discuss the use of 2D thermal recovery maps (TRMs) to compare the response to the stress of the samples and to detect in-layers structural damages. To the best of our knowledge, this is the first time that this approach based on TRMs is used to characterize composite materials and during structural inline tests in general. The thermal results achieved were compared with the stress analysis data acquired. Our results demonstrate how these analysis approaches allows a simple and rapid visualization of both the response of the materials subjected to inline stress and of their most damaged areas, thus representing effective methods to evaluate and compare the thermal and mechanical behaviors of newly developed composite materials.

## 2. Materials and Methods

### 2.1. Samples Realization 

The research was focused on composite laminated samples made from a commercial polymer blend supplied by Enyax s.r.l. (Milan, Italy), under the trade name A500, and reinforcements represented by fiber fabrics. Specifically, the polymeric phase A500 is a blend of polylactic acid (PLA) and polybutylene terephthalate-co-adipate (PBAT) combined in a ratio of 20/80 by weight, filled with about 12 wt% of microsized calcium carbonate (CaCO_3_) particles and, usually, intended for food packaging applications. Some of its main properties are: M.F.R.@190 °C/2.16 kg/6.0–8.0 g/10 min and density = 1.29 g/cm^3^.

The reinforcements taken into consideration, instead, are:A jute fiber fabric plain weave type, with an areal density of 290 g/m^2^ and supplied by Composite Evolution Ltd. (Chesterfield, UK);A plain weave basalt fabric with an areal weight of 210 g/m^2^ from Incotechnology GmbH (Pulheim, Germany).

Both samples were obtained following a process divided into two stages:Filming of the A500 blend, preliminarily dried in a vacuum oven at 70 °C overnight, with the aid of a flat die extruder Teach-Line E 20-T equipped with a calender CR 72T Collin (Ebersberg, Germany). The process conditions applied were: a temperature profile of 165°–175°–180°–170°–165° from the hopper to the die, and a screw speed of 60 rpm. Operating in this way, the matrix was transformed in a 100 μm-thick film.Film stacking and hot pressing: sample plates were prepared by alternately superimposing A500 film and fiber fabric layers and then consolidated by hot pressing using a lab-press Collin P400E (Ebersberg, Germany) at 180 °C, applying a pre-optimized pressure cycle. Regardless of the nature of the fibers, the process conditions adopted made it possible to obtain laminates with a thickness of approximately 2 mm.

From the two samples obtained, A500-Basalt and A500-Jute, rectangular specimens of 85 × 10 mm^2^ were cut from the composite plates for the subsequent infrared image analysis of the same, simultaneously subjected to flexural stress. For each composite sample, the analysis was performed on 5 specimens to ensure the reproducibility the reproducibility of the thermographic measurements performed. Specifically, the IR analysis, the experimental details of which will be specified in the following paragraph, was performed while each specimen was subjected to a three-point flexural test with the aid of a universal dynamometer (Instron Mod. 4505) equipped with at 1 kN load cell. The strain rate was set at 5 mm/min. In Figure 1, pictures of the raw materials and of the composite specimens fabricated are shown. 

### 2.2. Inline Infrared Imaging: Set-Up and Measurements 

Infrared imaging measurements were performed to characterize the thermal behavior of the materials investigated. For the measurements, a MWIR Camera FLIR X6580 sc (Winsonville, OR, USA) with a cooled indium antimonide (InSb) sensor, a spectral range of 3.5–5 μm, an FPA of 640 × 512 pixels, and a NETD of ~20 mK at 25 °C was used. The analysis was made using a germanium objective with a focal length of 50 mm and an IFOV of 0.3 mrad. The ResearchIR (FLIR Systems Inc., Winsonville, OR, USA) software was used to record the thermal images and for the basic analysis operations. The measurements were carried out during an inline quasi-static flexural test following two different approaches: passive and active. First, passive measurements were carried out by recording a temporal sequence of thermal images of the specimen under analysis, while the latter was progressively subjected to a bending, therefore, without the use of any external thermal stimulus. Subsequently, active analysis was performed after the specimen was subjected to bending for 240 s and without interrupting the stress test. It was thermally stimulated with a flash-lamp, and thermal images were recorded just before, during, and for 1 s after the pulse. Spatial TRMs of specimens were calculated by a specially designed MATLAB code (R2019b, Math-Works). In both passive and active analysis, the infrared images were recorded with a frame rate of 60 Hz. In the case of active analysis, the value of 60 Hz allows the sample to be considered stationary, as explained in Section 3.2. The flash-lamp system used was a Zoom Action head (Elinchrom, Renens, Switzerland) supported by a Digital 2400 RX power generator (Elinchrom) that allows light pulse of about 300 ms and 2400 J. In both approaches, the measurements were performed with the camera and the thermal source positioned on the same side of the specimen. The camera was positioned to simultaneously view the bottom surface and the thickness of the samples. Surface emissivity values of 0.96 for A500-Basalt specimens and 0.90 for A500-Jute specimens were estimated considering a black reference with an emissivity of 1.00 and set during the acquisition of thermal images. The measurements were made in the laboratory at a temperature of 22 °C and a humidity of 58%. Figure 2a,b, respectively, show a scheme and an image of the experimental set-up, while the images of one of the samples analyzed in Figure 2c,d are, respectively, at the beginning and at the end of the quasi-static flexural test.

## 3. Results and Discussion

The present work concerns the use of infrared imaging in the characterization of two different green composite materials during inline quasi-static flexural tests. The analysis was performed following two different approaches of this technique based on monitoring the response of the specimen without the latter being subjected to any external thermal stimulus (passive approach) and on analyzing its behavior in correspondence of an external thermal excitation caused by the use of a flash lamp (active approach). In this last case, 2D thermal recovery maps (TRMs) of the specimens under investigation were calculated and used to detect in-layer structural damage.

The purpose of these experimental evaluations was to highlight the contribution of infrared thermography to compare the different behaviors of these materials under inline stress tests and to show the potential of the TRMs to compare the areas of the specimens most affected by damages in the case of active analysis. In the following, the results obtained with the two different methods employed are shown and discussed in two separate sections.

### 3.1. Infrared Imaging: Passive Approach

Specimens of the two investigated materials A500-Basalt and A500-Jute were subjected to quasi-static flexural carried out with a standard three-point testing configuration, shown in Figure 2. During the test, the upper central nose (A in Figure 2c) present in the set-up remains fixed and stationary, while the two lower lateral noses (B and C in Figure 2c) move upwards, causing a gradually increasing stress on the specimen. Both the mechanical and thermal response to stresses of the specimens were recorded as a function of time from the start of the test for a time of 240 s. Thermal images were acquired throughout the span with a frame rate of 60 Hz. The synchronization of the thermal image acquisitions with the start of the mechanical test was performed manually. However, the high recording frame rate (60 Hz) compared to the slow bending speed (5 mm/min) makes the error in choosing a common start time negligible.

Considering the gradual deformation to which the specimen is subjected during the quasi-static test, the choice of an area from which to extrapolate thermal information from the sequence of frames acquired is certainly not trivial. The specimen surface changes its spatial position while it bends, and the coordinates of its pixels vary over time. This makes it impossible to easily and directly extrapolate the temperature trend in all pixels or a large area. In the literature, some post-processing methods are proposed based on home-made calculation procedures to remedy this problem [30]. However, due to the simultaneous deformation as well as the displacement of the specimen, these methods can only partially solve the problem in the case of analysis performed on the whole specimen. Furthermore, their use introduces further complexities and slowdowns in the data processing phase. To overcome this problem, in this study, the average temperature of an area (about 200 pixels) corresponding to the bottom layer of the specimens in proximity of the contact with the central fixed nose was chosen for the thermal monitoring (area S_1_ delimited by the red rectangle in the inset of Figure 2c). In fact, since this area is close to the bending rotation center and, therefore, subject to the minimum displacement (almost zero) during the test, the corresponding points of the specimen remain substantially the same, and the error made in calculating the average temperature is practically negligible. Before carrying out any subsequent analysis, relative only to the pixels of the area S_1_ taken into consideration, the temperature recorded in the first image (acquired at *t* = 0) was subtracted from all the thermal frames of the acquired sequence in order to consider only the temperature variations (Δ*T*) induced by the mechanical stress. This calculation is performed for all pixels of area S_1_ through the following basic operation:Δ*T*(*x*, *y*, *t*) = *T*(*x*, *y*, *t*) − *T*(*x*, *y*, 0)(1)
where *x* and *y* are the spatial coordinates in the thermal images and t in the temporal one. It should be noted that considering the temperature variations Δ*T*(*x*, *y*, *t*), rather than the absolute temperatures *T*(*x*, *y*, *t*) detected, allows us to reduce the errors due to an inexact knowledge of the emissivity of the matrices of the new materials investigated as well as to the influence of the environmental conditions. Before analyzing the results achieved, we recall here that two main different effects can be detected associated with their thermal response in the case of matrix-based composites: the thermoelastic and the thermoplastic effect. The first is associated with cooling due to the elastic expansion of the material and, in this case, the mechanical stress to which the specimen is subjected does not cause permanent variations in the shape or volume of it. The second, on the other hand, is associated with an increase in temperature due to the mechanical energy absorbed by the material and produces permanent modifications such as breakage, cracks, and delamination. As an example of the results obtained on the investigation of the two different materials, Figure 3 shows the graphs relating to the mechanical and thermal response recorded on an A500-Basalt specimen (Figure 3a) and one on an A500-Jute (Figure 3b). In both graphs, the mechanical response is shown as a solid black line and the thermal response (average temperature of the area S_1_) as a dashed black line. In order to improve visualization and reduce noise in thermal patterns, a smoothing of the latter measurements by adjacent averaging operation was performed and reported with a continuous red line in both graphs.

In the case of the A500-Basalt specimen (Figure 3a), the stress curve displays a rise until its maximum of about 24.4 MPa reached after 44.4 s and is accompanied in this phase by a cooling of the monitored S_1_ area that can be associated with a thermoelastic effect and, therefore, to non-permanent elastic deformation. The maximum of the stress curve coincides with the transition from the thermoelastic to the thermoplastic effect and, therefore, with the beginning of the generation of permanent damages to the specimen starting from its layers close to the upper surface in contact with the pushing nose. The minimum Δ*T* value of about −0.35 K (red line) is recorded after 45.9 s, 1.5 s after the maximum stress. This delay between the two signals can be due to the time that the heat released in the generation of damage spent to travel from the top surface of the specimen (directly in contact with the load) to the bottom surface where the thermal signal is recorded. After the maximum value, the stress curve shows a gradual decrease, characterized by some almost constant stretches. This gradual decrease and the absence of an abrupt drop in the signal indicates that the specimen fibers slowly wear out under mechanical stress without showing a real collapse but rather a linear deformation. In this phase, the area S_1_ shows a quasi-linear temperature increase, with Δ*T* reaching up to 0.53 K (with an excursion from its minimum of 0.88 K), which can be attributed to the thermoplastic effect and to the heat generated in the formation of damages and permanent delaminations between the layers of the specimen. The small Δ*T* peaks visible in the thermal response curve (dashed black line) can be attributed to the sequential formation of mini-cracks in the specimen and to the friction effect between its various layers.

In the case of the A500-Jute specimen (Figure 3b), both the stress and the thermal curve show a trend similar to the previous case but with some relevant differences. The stress curve for A500-Jute rises to a maximum of about 21.7 MPa, similar to A500-Basalt, but reaches this value only after 83.2 s, thus spending almost double the time, indicating a greater resilience of this type of matrix. This phase is accompanied by a cooling of the S_1_ area characterized by a Δ*T* of about −0.13, lower than the A500-Basalt specimen, indicating that the A500-Jute matrix is less affected by non-permanent elastic deformation. In this case, the delay between the minimum Δ*T* and the maximum stress is about 2.9 s, about double the time compared to the previous case, reasonably, due to the higher thermal conductivity of the jute fibers (about 0.4 W m^−1^ K^−1^) [31] compared to the basalt fibers (about 0.04 W m^−1^ K^−1^) [32], which favors the horizontal dispersion of heat along the layers of the A500-Jute specimen, decreasing the amount of energy that propagates vertically from the upper layers to the lower ones. Additionally, in the case of the A500-Jute, after the stress signal has reached the maximum value, no abrupt drops are evident, and the signal decreases slowly, indicating how the permanent damages that characterize this thermoplastic phase are gradually generated in the layers of the specimen. The S_1_ area shows a quasi-linear temperature increase with Δ*T* reaching up to 0.63 K, with an excursion from its minimum of 0.76 K, slightly lower than the case of the basalt specimen, indicating that the total energy released in the damage generation for the two materials investigated is quite similar.

### 3.2. Infrared Imaging: Active Approach

Specimens of the two materials taken into account, A500-Basalt and A500-Jute, were analyzed using an active infrared imaging approach during the quasi-static flexural test. The analysis was performed without interrupting the test and after the specimens were subjected to flexural stress for 240 s. The purpose of these measurements was to verify the possibility of detecting stress damage on the specimens directly during the test. The area taken into consideration for this analysis is indicated by a red rectangle in Figure 2d (Area S_2_, 185 × 148 pixels). Samples were heated through the use of a flash-lamp, and their thermal images were recorded just before, during, and for 1 s after the pulse with a frame rate of 60 Hz. As an example, Figure 4 shows some of the thermal images extrapolated at different times from the sequences recorded for two of the characterized specimens: the last frame before the pulse (*t* = 0^−^), the first unsaturated frame after the pulse (*t* = 0^+^), and the frames acquired after 0.25 s, 0.50 s, and 1.0 s from time *t* = 0^+^.

It should be noted that the different temperatures recorded for the metallic elements (circle/cylinder/roller) present outside the investigated specimens represent an artifact due to the different emissivity values (reported in Section 2.2), with which these two thermal analyses were conducted.

To detect damage and compare the state of the specimens, we calculated their TRMs by analyzing the temporal trend in the temperature from the frames acquired. For this purpose, as a first step, in order to consider only the temperature variations Δ*T* from all the thermal frames acquired after the pulse (*t* ≥ 0^+^), the temperature *T*(*x*, *y*, 0^−^) recorded in the last image acquired before the pulse (*t* = 0^−^) was subtracted:Δ*T*(*x*, *y*, *t* ≥ 0^+^) = *T*(*x*, *y*, *t* ≥ 0^+^) − *T*(*x*, *y*, 0^−^)(2)

The first image of this sequence represents the thermal gap induced Δ*T_induced_*(*x*, *y*, *t* = 0^+^) for each pixel by the flash lamp excitation. Subsequently, the TRMs were obtained by calculating the time it takes to recover 80% of the specific Δ*T_induced_*(*x_p_*, *y_p_*, *t* = 0^+^) from the temporal trend in each pixel (*x_p_*, *y_p_*). According to our results, the 80% threshold represents the value that allows us to obtain a greater contrast between the areas affected by damage and the intact areas. However, by choosing other threshold values in the range of 75–85%, the results of the analysis were not substantially affected. It is important to note that the high frame rate of image recording (60 Hz) compared to the slow movement of the lateral noses (5 mm/min) allowed us to consider the specimen stationary and, therefore, not to make errors in the evaluation of the recovery times of the pixels, although the images were recorded during the execution of the test, therefore, without interrupting the movement of the specimen. In fact, in the calculation, the temporal trend in the pixels was obtained from 60 images acquired in 1 s; in this time interval, the displacement of each point of the specimen was about 80 μm well below the spatial resolution associated with each pixel of the images, equal to about 260 μm. It should be noted that the latter condition of the stationarity of the sample during the analysis cannot be easily satisfied with the application of other notable active approaches present in the literature, such as lock-in thermography (LiT) [33,34] or pulsed phase thermography (PPT) [35]. These latter techniques require much longer time for heating and image recording than the method proposed here, making them difficult to apply for inline analysis where the position of the sample varies over time.

As an example, Figure 5 shows the TRMs calculated for two specimens characterized (some thermal images of which are shown in Figure 4), respectively, of A500-Basalt (Figure 5a) and A500-Jute (Figure 5b). In the maps, the background was removed through a homemade MATLAB procedure to improve their visualization.

Both maps highlight areas of the specimens characterized by different recovery times. To give an interpretation of the maps calculated in relation to the presence of damage in the specimens, the following observation can be made. The areas most affected by subsurface damage, such as the delamination between the layers of the specimen, mini-cracks, and internal breaks, are characterized by the presence of micro air gaps generated in correspondence with them. Due to their greater thermal inertia, the micro air gaps slow down the heat diffusion process and induce longer heat recovery times. Therefore, through this consideration, it is possible to match the recovery time shown on the map by an area of the specimen with its level of damage. In particular, the areas with longer recovery times are those most affected by structural damage, while those corresponding to shorter recovery times are less influenced and vice versa. In Figure 5c,d, the areas of the two specimens most affected by damage characterized by recovery times longer than 0.8 s are highlighted in dark red. 

In the case of the TRM of A500-Basalt (Figure 5a,c), the stress tests induced structural damage along the whole thickness of the specimen. The most affected area is the one in contact with the central nose, where the effects of the damage reach the layers close to the bottom surface. Furthermore, as can be seen from the map, even the first sections of the more superficial layers were affected by the stresses applied.

In comparison, the TRM of the A500-Jute (Figure 5b,d) shows damages localized mainly in the more superficial layers of the specimen whose integrity was compromised along their whole length. The deeper layers near the bottom surface do not seem particularly affected by the applied stress.

For both specimens, the percentages of the area mainly affected by damages were determined using the following formula, *N_DAM_*/*N_SAM_* × 100, where *N_DAM_* and *N_SAM_* represent the number of pixels associated with damages (dark red pixels) and the total number of pixels of the samples (dark red pixels + light blue pixels), respectively, in the TRM reported in Figure 5c,d. From these calculations, 25.9% of the area of the A500-Basalt specimen was most affected by damage, while this percentage drops to only 14.6% for the A500-Jute specimen. It should be pointed out that this method can be carried out and repeated several times during the inline test, thus allowing us to monitor the response of the samples to the different loads. These results show how the active infrared imaging approach considered here, based on the use of TRMs, represents a technique of certain interest for the analysis and characterization of new materials subjected to inline tests such as the one considered in this work.

Compared to other applications of infrared imaging in inline quasi-static flexural testing proposed in past years [12,25,30], the method discussed here provides a direct visualization of the areas mainly affected by damage, allowing for a rapid estimation of the size of the affected area. Furthermore, no other evidence of the application of active infrared approach during inline tests on composite materials is present in the literature. 

However, two main limitations of the analytical approach shown can be highlighted. The proposed method allows us to identify the areas of the investigated samples most affected by damage, allowing for a quantitative evaluation of the percentage of the damaged area. However, the method does not allow for differentiation between the possible damages (crack, delaminations, layer detachments, and others) that the stress force to which the specimens are subjected can produce. To overcome this limitation, a more complex and accurate calculation procedure (but that also requires more processing times) must be implemented starting from a more in-depth and precise knowledge of the physical–thermal characteristics of the investigated samples. Another critical but important point is represented by the high costs of the infrared camera typology (MWIR with cooled sensor), with which the methods discussed here have been tested and implemented. Future comparison analyses between the results obtainable with these types of infrared camera and other lower cost ones also operating in the LWIR spectral range must be performed to understand the limits in performance of the proposed approach of analysis. Moreover, the implementation of the infrared imaging analysis proposed with a low-cost camera would further increase their interest for industrial applications.

## 4. Conclusions

In summary, in this work, fully eco-friendly laminated composites including woven jute and basalt fibers were investigated by the inline monitoring of three-point flexural tests with infrared thermography. The results of analysis achieved by Infrared imaging approaches performed in both passive and active ways during the inline quasi-static flexural test are compared with those obtained by mechanical stress. Passive infrared analysis allows us to distinguish between the thermoelastic and the thermoplastic effect induced on the materials investigated by the different phases of the test. Active infrared analysis based on the calculation of the TRMs allows us to detect the most damaged areas of the specimens during the test without the need to interrupt it. Both characterization approaches allow us to detect and compare the behavior of the investigated materials, highlighting their differences. The information provided by the techniques can help to understand the mechanical performance and limits of the investigated materials and can suggest possible improvements. These findings suggest that the characterization approaches shown are of interest for the analysis and development of new generation composite material matrices and for their comparison when subjected to inline tests. Furthermore, the simplicity of use makes this imaging analysis available also in industrial environments, allowing a non-destructive test, which can be performed simultaneously with other inline characterizations and is controllable remotely without requiring process interruption.

## Figures and Tables

**Figure 1 materials-16-03081-f001:**
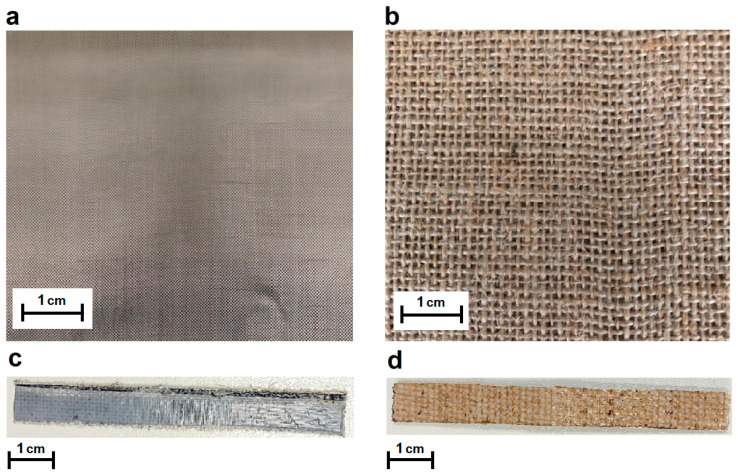
Pictures of the raw materials used: plain weave basalt fabric (**a**) and plain weave jute fabric (**b**). Pictures of the specimens fabricated: A500/Basalt (**c**) and A500/jute (**d**).

**Figure 2 materials-16-03081-f002:**
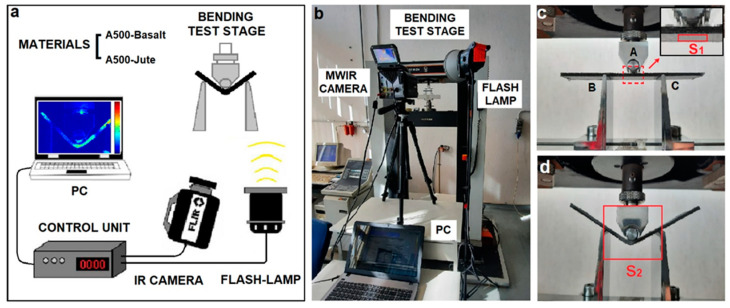
Experimental set-up for infrared imaging analysis during inline quasi-static flexural test: scheme of the experimental set-up (**a**), image of the experimental set-up (**b**), and image of a specimen at the beginning (**c**) and at the end (**d**) of the test. In (**c**) A indicates the central fixed nose while B and C the lateral dynamic ones. In (**c**) S_1_ indicate the area of the specimens considered in the passive analysis while in (**d**) S_2_ the area considered in the active approach.

**Figure 3 materials-16-03081-f003:**
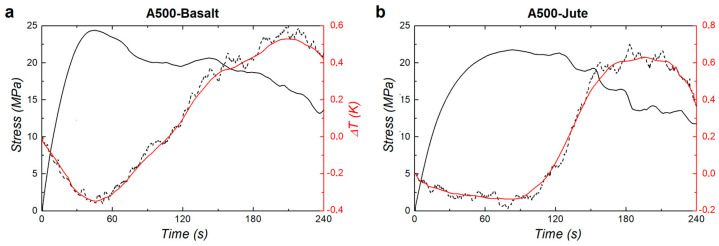
Examples of mechanical and thermal response recorded on the samples analyzed: A500-Basalt (**a**) and A500-Jute (**b**).

**Figure 4 materials-16-03081-f004:**
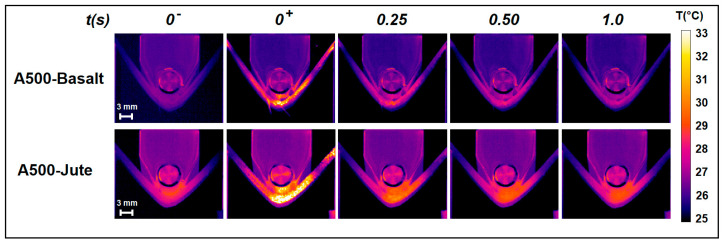
Thermal images extrapolated at different time from the sequences recorded for the two specimens A500-Basalt (first row) and A500-Jute (second row): the last frame before the pulse (*t* = 0^−^), the first unsaturated frame after the pulse (*t* = 0^+^), and the frames acquired after 0.25 s, 0.50 s, and 1.0 s from time *t* = 0^+^.

**Figure 5 materials-16-03081-f005:**
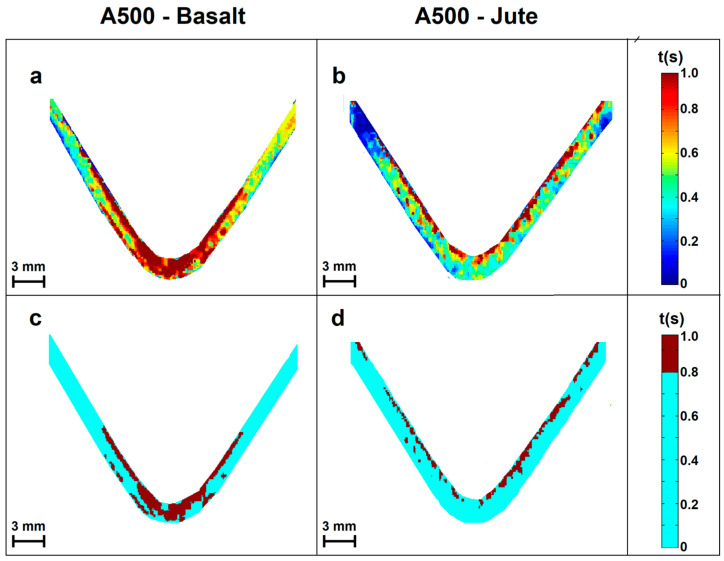
TRMs calculated for two specimens: A500-Basalt (**a**) and A500-Jute (**b**). The area of the two specimens characterized by recovery times longer than 0.8 s are shown in dark red in (**c**) and in (**d**).

## Data Availability

All data that support the findings of this study are included within the article.

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
