# Peer review of "Infrared Imaging Analysis of Green Composite Materials during Inline Quasi-Static Flexural Test: Monitoring by Passive and Active Approaches"

_materials, 2023, doi:10.3390/ma16083081_

Round 1
Reviewer 1 Report
for abstract and conclusion, if can add the findings, it will be good (highlight the results)
line 336 & 337: how did you determine this area?show the equation/formula
based on this percentage area, how to determine either the damage is insignificant or greatly significant?
is it possible to determine thermal conductivity using the active approach?
line 298 60 images were used to determine the temporal trend of the pixels, is it standard calculation is based on 60 images?please state clearly in the methodolgy.
Author Response
Please see the attachment.
Massimo Rippa

Reviewer 2 Report
General Comments to the Authors
General purpose and concept of this manuscript titled “Infrared imaging analysis of green composite materials during inline quasi-static flexural test: monitoring by passive and active approaches” seems to be appropriate for“Materials”. The authors produced jute/basalt-based composites. The authors determined flexural tests and IR analysis. However, the authors should enhance quality of the manuscript before publication.
Specific comments are as follows:
The manuscript should be revised by a native English speaker.
More literature search should be done on lignocellulosic materials reinforced composites and summarize in the Introduction section of this manuscript.
It is highly recommended that the authors should read and summarize the below articles into the manuscript.
ü Merzoug, A., Bouhamida, B., Sereir, Z., Bezazi, A., Kilic, A., Candan, Z. 2020. Quasi-static and dynamic mechanical thermal performance of date palm/glass fiber hybrid composites. Journal of Industrial Textiles.
ü Candan, Z., Tozluoglu, A., Gonultas, O., Yildirim, M., Fidan, H., Alma, M.H., Salan, T. 2022. Nanocellulose: Sustainable biomaterial for developing novel adhesives and composites. Industrial Applications of Nanocellulose and Its Nanocomposites. Elsevier, UK, pp. 49-137.
Homogeneity of the matrix?
Photographs of the raw materials and the composites should be added.
How many samples did the authors use for each test?
Statistical analysis for mechanical properties should be performed and added into the manuscript.
Some conclusions and suggestions regarding with industrial perspective should be added into the Conclusions section of the manuscript.
Other mechanical and thermal properties of the composites should be supplied.
Author Response

(The authors gave the same response as above.)

Reviewer 3 Report
This paper “Infrared imaging analysis of green composite materials during inline quasi-static flexural test: monitoring by passive and active approaches” studies the behaviour of totally eco-friendly composite materials during flexural tests through infrared (IR) analysis. IR imaging is a well-known non-contact technique and represents a reliable means of providing low-cost in situ analysis. According to this method, the surface of the sample under investigation is monitored, under natural conditions or after heating, by recording thermal images with an appropriate IR camera. Authors reported and discussed their results achieved for jute and basalt based eco-friendly composite using both passive and active IR imaging approaches.
The topic is justified. The paper could be further improved if the following remarks are taken into consideration:
1. ABSTRACT: The text should include more details about the proposed methodology, numerical results achieved, and a comparison with other methods (if possible).
2. Few grammatical mistakes and extra spaces before the start of sentence were found in the draft of the article.
3. Introduction section lacks a proper introduction of the whole of the conducted research, background, justification of the research, and major contributions of the study. The contribution may be key fold in the introduction section.
4. At-least paper should add one base paper, as a reviewed literature and for comparison purpose.
5. Results and discussion section only presents the information in connection with results only, there is no proper discuss, as it could not be without comparison.
6. The motivation is not clear. Please specify the importance of the Infrared imaging analysis.
7. Discuss the limitations of the proposed method with their possible solutions in the future work section.
Author Response
Please see the the attachment.
Massimo Rippa

Reviewer 4 Report
Review of “Infrared imaging analysis of green composite materials during inline quasi-static flexural test: monitoring by passive and active approaches,” by Massimo Rippa, Vito Pagliarulo, Francesco Napolitano, Teodoro Valente, and Pietro Russo
Paper describes an interesting experimental investigation of damage assessment and thermal changes of new ‘green’ material samples subject to bending (flexing) stresses. The experiment is described following two methodologies using MWIR infrared camera to image the samples during the actuation pressure imposed by pushing the sample center with what looks like a roller cylinder. In one approach only camera (passive) is used and in another a high-power light source pulses to heat the samples (active) is added. Thermal recovery maps are drawn from 60 Hz camera rate interrogating a slowly flexing material. Data image processing is designed to account for pixel motion and resolutions.
Paper should be of interest to readers in the field and it is recommended for publication after the following clarifications and concerns are addressed satisfactorily.
Needs clarification of how mechanical stress is measured (such as in Fig. 2 data)
Temperature difference seems allows using IR measurements without knowing emissivity; however, how the Temperature maps without emissivity values (Fig. 3) are obtained should be clarified: ‘how is temperature measured without knowing emissivity’
Explain why choice of MWIR; for low temperatures a LWIR might be preferred, if available
Consider validating temperature from the IR camera with secondary measurement, such as thermocouples to provide a calibration of the IR imaging; maybe regions far from the stress points can be used for that.
The LED light is probably broadband and have some MWIR components that potentially reflect off the surface; in the active method the light radiation might produce a reflection from the object that the camera interprets as thermal radiation instead; need verify no reflections are imposed in the data.
Cracks and gaps are expected to have different emissivity from the sample body; need point out if this affects the data.
Surface and sub-surface defects are noted but how deep into the sample this can be measured is not clear.
Need clarify how the camera views the sample, is ambiguous if is the side or the bottom or both.
Need explain IR maps (Fig. 3) features that are observed outside the sample: around the circle/cylinder/roller that pushes the sample there is a dark circle around (perhaps explaining the air or contact area interface with the top trapezoidal fixture) but is not same for both samples (appears it should be same if is same fixture and roller used in both samples); also there are regions between the sample and the roller that seem as hot as the sample but is unclear what these regions are
Page 2 typo: missing micro symbol between ‘100’ and ‘m’
Page 8 1st paragraph line 324: seems ‘interested’ should be changed to another word.
If possible analytical tools such as Finite Element Analysis FEA of the sample would be good complementary information to the experiments.
Author Response

(The authors gave the same response as above.)
